# Multi-scale Hierarchical Vision Transformer with Cascaded Attention Decoding for Medical Image Segmentation

**Md Mostafijur Rahman**                                    MOSTAFIJUR.RAHMAN@UTEXAS.EDU

**Radu Marculescu**                                              RADUM@UTEXAS.EDU

*System Level Design Group, Department of ECE, The University of Texas at Austin*

**Editors:** Accepted for publication at MIDL 2023

## Abstract

Transformers have shown great success in medical image segmentation. However, transformers may exhibit a limited generalization ability due to the underlying single-scale self-attention (SA) mechanism. In this paper, we address this issue by introducing a Multi-scale hiERarchical vIsion Transformer (MERIT) backbone network, which improves the generalizability of the model by computing SA at multiple scales. We also incorporate an attention-based decoder, namely Cascaded Attention Decoding (CASCADE), for further refinement of the multi-stage features generated by MERIT. Finally, we introduce an effective multi-stage feature mixing loss aggregation (MUTATION) method for better model training via implicit ensembling. Our experiments on two widely used medical image segmentation benchmarks (i.e., Synapse Multi-organ and ACDC) demonstrate the superior performance of MERIT over state-of-the-art methods. Our MERIT architecture and MUTATION loss aggregation can be used with other downstream medical image and semantic segmentation tasks.

**Keywords:** Medical image segmentation, Vision transformer, Multi-scale transformer, Feature-mixing augmentation, Self-attention.

## 1. Introduction

Automatic medical image segmentation has become an important step in disease diagnosis nowadays. Since the emergence of UNet (Ronneberger et al., 2015), U-shaped convolutional neural networks (CNNs) (Oktay et al., 2018; Huang et al., 2020; Zhou et al., 2018; Fan et al., 2020) have become de facto methods for medical image segmentation. By producing high-resolution segmentation maps through aggregating multi-stage features via skip connections, UNet variants, such as UNet++ (Zhou et al., 2018) and UNet3Plus (Huang et al., 2020), have shown good performance in medical image segmentation. However, the spatial context of the convolution operation limits the ability of CNN-based methods to learn the long-range relations among pixels (Cao et al., 2021). Some work (Chen et al., 2018; Oktay et al., 2018; Fan et al., 2020) try to address this issue by embedding attention mechanisms in the encoder or decoder. However, despite the significant efforts made in this direction, the CNN-based methods still have insufficient ability to capture long-range dependencies.

With the emergence of Vision transformers (Dosovitskiy et al., 2020), many works (Cao et al., 2021; Chen et al., 2021; Dong et al., 2021; Wang et al., 2022b) try to address the above problem using a transformer encoder, specifically designed for medical image segmentation. Transformers capture long-range dependencies among pixels by learning correlations among all the input patches using self-attention (SA). Recently, hierarchical vision transformers,

such as pyramid vision transformer (PVT) (Wang et al., 2021) with spatial reduction attention, Swin transformer (Liu et al., 2021) with window-based attention, and MaxViT (Tu et al., 2022) with multi-axis attention have been introduced to improve performance. Indeed, these hierarchical vision transformers are very effective for medical image segmentation tasks (Cao et al., 2021; Dong et al., 2021; Wang et al., 2022b). However, these transformer-based architectures have two limitations: 1) self-attention is performed with a single attention window (scale) which has limited feature processing ability, and 2) self-attention modules used in transformers have limited ability to learn spatial relations among pixels (Chu et al., 2021).

More recently, PVTv2 (Wang et al., 2022c) embeds convolution layers in transformer encoders, while CASCADE (Rahman and Marculescu, 2023) introduces an attention-based decoder to address the limitation of learning spatial relations among pixels. Although these methods enable learning of the local (spatial) relations among pixels, they still have limited ability to capture features of multi-scale (e.g., small, large) organs/lesions/objects due to the single-scale attention window used to compute the self-attention. To address this limitation, we introduce a novel *multi-scale hierarchical* vision transformer (MERIT) backbone which computes self-attention across *multiple attention windows* to improve the generalizability of the model. We also incorporate multiple CASCADE decoders to produce better high-resolution segmentation maps by effectively aggregating and enhancing multi-scale hierarchical features. Finally, we introduce a novel effective multi-stage (hierarchical) feature-mixing loss aggregation (MUTATION) strategy for implicit ensembling/augmentation which produces new synthetic predictions by mixing hierarchical prediction maps from the decoder. The aggregated loss from these synthetic predictions improves the performance of medical image segmentation. Our contributions are as follows:

- **New Transformer Architecture:** We propose a new multi-scale hierarchical vision transformer (MERIT) for 2D medical image segmentation which captures both multi-scale and multi-resolution features. Besides, we incorporate a cascaded attention-based decoder for better hierarchical multi-scale feature aggregation and refinement.

- **Multi-stage Feature-mixing Loss Aggregation:** We propose a new simple, yet effective way, namely MUTATION, to create synthetic predictions by mixing features during loss calculation; this improves the medical image segmentation performance.

- **Better State-of-the-art Results:** We perform rigorous experiments and ablation studies on two medical image segmentation benchmarks, namely Synapse multi-organ and ACDC cardiac diagnosis. Our implementation of MERIT using two instances (with different windows for SA) of MaxViT (Tu et al., 2022) backbone with CASCADE decoder and MUTATION loss aggregation strategy produces new state-of-the-art (SOTA) results.

## 2. Related Work

### 2.1. Vision transformers

Dosovitskiy et al. (Dosovitskiy et al., 2020) build the first vision transformer (ViT), which can learn long-range (global) relations among the pixels through SA. Recent works focus on improving ViT in different ways, such as designing new SA blocks (Liu et al., 2021; Tu

et al., 2022), incorporating CNNs (Wang et al., 2022c; Tu et al., 2022), or introducing new architectural designs (Wang et al., 2021; Xie et al., 2021). Liu et al. (Liu et al., 2021) introduce a sliding window attention mechanism in the hierarchical Swin transformer. In DeiT (Touvron et al., 2021), authors explore data-efficient training strategies to minimize the computational cost for ViT. SegFormer (Xie et al., 2021) proposes a positional-encoding-free hierarchical transformer using Mix-FFN blocks. In PVT, authors (Wang et al., 2021) develop a pyramid vision transformer using a spatial reduction attention mechanism. The authors extend the PVT to PVTv2 (Wang et al., 2022c) by embedding an overlapping patch embedding, a linear complexity attention layer, and a convolutional feed-forward network. Recently, in MaxViT (Tu et al., 2022), authors propose a multi-axis self-attention mechanism to build a hierarchical hybrid CNN transformer.

Although vision transformers have shown excellent promise, they have limited spatial information processing ability; also, there is little effort in designing multi-scale transformer backbones (Lin et al., 2022). In this paper, we address these very limitations by introducing a multi-scale hierarchical vision transformer with attention-based decoding.

## 2.2. Medical image segmentation

Medical image segmentation can be formulated as a dense prediction task of classifying the pixels of lesions or organs in endoscopy, CT, MRI, etc. (Dong et al., 2021; Chen et al., 2021). U-shaped architectures (Ronneberger et al., 2015; Oktay et al., 2018; Zhou et al., 2018; Huang et al., 2020; Lou et al., 2021) are commonly used in medical image segmentation because of their sophisticated encoder-decoder architecture. Ronneberger et al. (Ronneberger et al., 2015) introduce UNet, an encoder-decoder architecture that aggregates features from multiple stages through skip connections. In UNet++ (Zhou et al., 2018), authors use nested encoder-decoder sub-networks that are linked using dense skip connections. Finally, UNet3Plus (Huang et al., 2020) explores the full-scale skip connections having intra-connections among the decoder blocks.

Transformers are nowadays widely used in medical image segmentation (Cao et al., 2021; Chen et al., 2021; Dong et al., 2021). In TransUNet (Chen et al., 2021), authors propose a hybrid CNN transformer architecture to learn both local and global relations among pixels. Swin-Unet (Cao et al., 2021) introduces a pure U-shaped transformer using Swin transformer (Liu et al., 2021) blocks. Recently, in CASTFormer (You et al., 2022), authors introduce a class-aware transformer with adversarial training.

Some studies explore attention mechanisms with CNN (Oktay et al., 2018; Fan et al., 2020) and transformer-based architectures (Dong et al., 2021) for medical image segmentation. In PraNet (Fan et al., 2020), authors utilize the reverse attention (Chen et al., 2018). PolypPVT (Dong et al., 2021) uses PVTv2 (Wang et al., 2022c) as the encoder and adopts a CBAM (Woo et al., 2018) attention block in the decoder with other modules. In CASCADE (Rahman and Marculescu, 2023), authors propose a cascaded decoder using attention modules for feature refinement. Due to its remarkable performance in medical image segmentation, we incorporate the CASCADE decoder within our architecture.

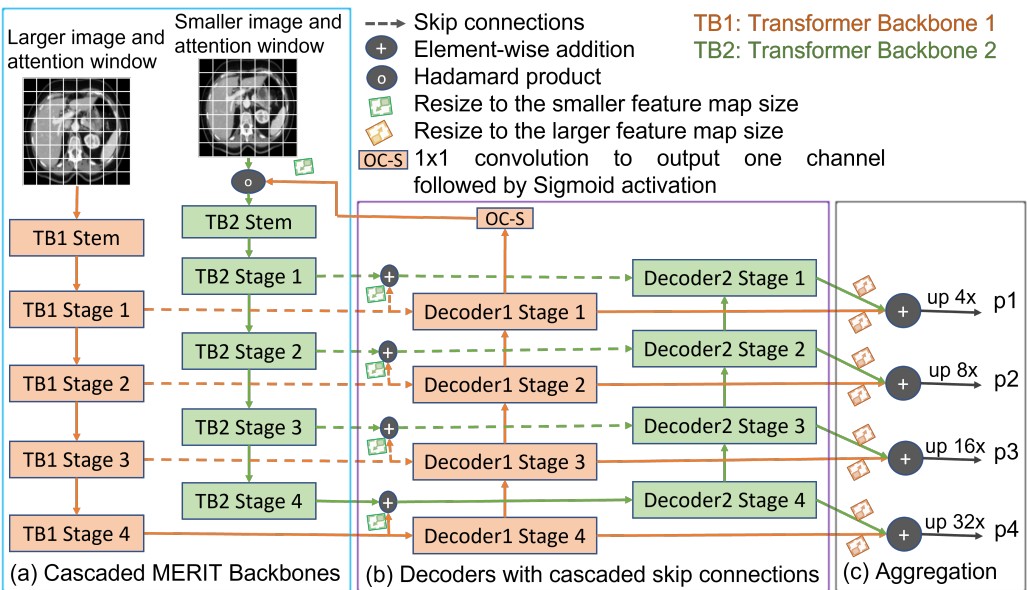

Figure 1: Cascaded MERIT architecture. (a) Cascaded MERIT backbone, (b) Decoders with cascaded skip connections from the decoder 1, (c) Prediction maps aggregation of two decoders. p1, p2, p3, and p4 are the aggregated multi-stage prediction maps.

## 3. Method

In this section, we first introduce our proposed multi-scale hierarchical vision transformer (MERIT) backbone and decoder. We then describe an overall architecture combining our MERIT (i.e., MaxViT (Tu et al., 2022)) with the decoder (i.e., CASCADE (Rahman and Marculescu, 2023)). Finally, we introduce a new hierarchical feature-mixing loss aggregation method.

### 3.1. Multi-scale hierarchical vision transformer (MERIT)

To improve the generalizability of the model across small and large objects in an image, we propose two designs (i.e., Cascaded and Parallel) based on the MERIT backbone network.

#### 3.1.1. CASCADED MERIT

In the cascaded design of our MERIT architecture, we add (i.e., cascade) feedback from a backbone to the next backbone. We extract the hierarchical features from four different stages of the backbone network. Then, we cascade these features with the features from the previous backbone and pass them to the skip connections and bottleneck modules of the respective decoders, except the first decoder. We also pass feedback from the decoder of one backbone to the next backbone, except the last. This design captures the multi-scale, as well as multi-resolution features due to using multiple attention windows and hierarchical features. It also refines the features well due to adding some feedback from the decoder of a backbone to the next backbone via cascaded skip connections. Fig. 1(a) presents the

Cascaded MERIT architecture with two backbone networks. For each backbone network, the images with size (H, W) are first put into a Stem layer (TB1 Stem, TB2 Stem in Fig. 1(a)) which reduces the resolution of the features to (H/4, W/4). Afterward, these features are passed through four stages of transformer backbones (this reduces the resolution of the features by 2 times at each stage, except the fourth). The features from the last stage of the first decoder are combined with the input image to cascade it with the second backbone in Fig. 1(a). To do this, we reduce the number of channels to one and produce logits by applying a $1 \times 1$ convolution followed by Sigmoid activation. We also resize the feature map to the input resolution (i.e., $224 \times 224$ in our implementation) of Backbone 2.

### 3.1.2. Parallel MERIT

Unlike Cascaded MERIT, in the parallel design of the MERIT backbone, we pass input images of multiple resolutions *in parallel* into separate hierarchical transformer backbone encoders with different attention windows. Similar to the Cascaded MERIT, we extract the hierarchical features from four different stages of the backbone networks and pass those features to the respective parallel decoders. This design also captures multi-scale features due to using hierarchical backbones with multiple attention windows. Fig. 2(a) in Appendix A presents a design for the Parallel MERIT with two backbone networks. The input images are passed through similar steps in the backbone networks, just as in the Cascaded MERIT. However, the Parallel MERIT shares information among the backbone networks only at the very end during the feature aggregation step (Fig. 2(c) in Appendix A).

## 3.2. Decoder

We propose using a separate decoder for each transformer backbone. As shown in Fig. 1(b), we use cascaded skip connections in the decoder of our cascaded MERIT architecture. Here, we add the skip connections from the first backbone to the skip connections of the second backbone network. In this case, we share information across backbones in three phases, i.e., during backbone cascading, skip connections cascading, and aggregating prediction maps. This sharing of information helps to capture richer information than the single-resolution backbone, as well as our Parallel MERIT.

Unlike Fig. 1(b), in Fig. 2(b) in Appendix A, we have two parallel decoders for our parallel backbones. Each decoder has four stages that correspond to four stages of the transformer backbone. We only aggregate the multi-stage prediction maps produced by the decoders in Fig. 2(b) at the aggregation step shown in Fig. 2(c).

## 3.3. Overall Architecture

In our experiments, we use one of the most recent SOTA transformers, MaxViT (Tu et al., 2022). We use two instances of MaxViT-S (standard) backbone with $8 \times 8$ and $7 \times 7$ attention windows to create our MERIT backbone. Each MaxViT backbone has two Stem blocks followed by four stages that consist of multiple (i.e., 2, 2, 5, 2) MaxViT blocks. Each MaxViT block is built with a Mobile Convolution Block (MBConv), a Block Attention having Block Self-Attention (SA) followed by a Feed Forward Network (FFN), a Grid Attention having a Grid SA followed by an FFN. We note that although we use the MaxViT backbone in our experiments, other transformer backbones can easily be used with our MERIT.

Pure transformers have limited (spatial) contextual information processing ability among pixels. As a result, the transformer-based models face difficulties in locating discriminative local features. To address this issue, we adopt a recent attention-based cascaded decoder, CASCADE (Rahman and Marculescu, 2023), for multi-stage feature refinement and aggregation. CASCADE decoder uses the attention gate (AG) (Oktay et al., 2018) for cascaded feature aggregation and the convolutional attention module (CAM) for robust feature map enhancement. CASCADE decoder has four CAM blocks for the four stages of hierarchical features from the transformer backbone and three AGs for three skip connections. CASCADE decoder aggregates the multi-resolution features by combining the upsampled features from the previous stage of the decoder with the features from the skip connections using AG. Then, CASCADE decoder processes the aggregated features using the CAM module (consists of channel attention (Hu et al., 2018) followed by spatial attention (Chen et al., 2017)) which groups pixels together and suppresses background information. Lastly, CASCADE decoder sends the output from the CAM block of each stage to a prediction head to produce prediction maps.

We produce four prediction maps from the four stages of the CASCADE decoder. As shown in Figs. 1(c) and 2(c) in Appendix A, we aggregate (add) the prediction maps for each stage of our two decoders. We generate the final prediction map, $\hat{y}$, using Equation 1:

$$\hat{y} = \alpha \times p1 + \beta \times p2 + \gamma \times p3 + \psi \times p4 \tag{1}$$

where $p1$, $p2$, $p3$, and $p4$ represent the prediction maps, and $\alpha$, $\beta$, $\gamma$, and $\psi$ are the weights of each prediction heads. We use the value of 1.0 for $\alpha$, $\beta$, $\gamma$, and $\psi$. Finally, we apply Softmax activation on $\hat{y}$ to get the multi-class segmentation output.

### 3.4. Multi-stage feature-mixing loss aggregation (MUTATION)

We now introduce a simple, yet effective multi-stage feature mixing loss aggregation strategy for image segmentation, which enables better model training. Our intention is to create new prediction maps by combining the available prediction maps. So, we take all the prediction maps from different stages of a network as input and aggregate the losses of prediction maps generated using $2^n - 1$ non-empty subsets of $n$ prediction maps. For example, if a network produces 4 prediction maps, our multi-stage feature-mixing loss aggregation produces a total of $2^4 - 1 = 15$ prediction maps including 4 original maps. This mixing strategy is simple, as it does not require additional parameters to calculate, and it does not introduce inference overheads. Due to its potential benefits, this strategy can be used with *any* multi-stage image segmentation or dense prediction networks. Algorithm 1 presents the steps to produce new prediction maps and loss aggregation.

## 4. Experiments

In this section, we demonstrate the superiority of our proposed MERIT architectures by comparing the results with SOTA methods. We introduce datasets, evaluation metrics, and implementation details in **Appendix B**. More experiments and ablation studies to answer questions related to our architectures are given in **Appendix C.1-C.7**.

---

**Algorithm 1:** Multi-stage Feature-Mixing Loss Aggregation

---

**Input:** $y$; the ground truth mask

A list $[P_i]$; $i = 0, 1, \cdots, n-1$, where each element is a prediction map

**Output:** $loss$; the aggregated loss

**1** $loss \leftarrow 0.0$;

**2** $\mathcal{S} \leftarrow$ find all non-empty subsets of prediction map indices, $\{0, \ldots, n-1\}$; // $\mathcal{S}$ is the set of non-empty subsets of $\{0, \ldots, n-1\}$

**3 foreach** $s \in \mathcal{S}$ **do**

**4**   $\hat{y} \leftarrow 0.0$; // $\hat{y}$ is a new prediction map

**5**   **foreach** $i \in s$ **do**

**6**     $\hat{y} \leftarrow \hat{y} + P_i$;

**7**   **end**

**8**   $loss \leftarrow loss\_function(y, \hat{y})$; // $loss\_function(.)$ is any loss function (e.g., CrossEntropy, DICE)

**9 end**

---

Table 1: Results on Synapse multi-organ dataset. DICE scores (%) are reported for individual organs. The results of UNet, AttnUNet, PolypPVT, and SSFormerPVT are taken from CASCADE (Rahman and Marculescu, 2023). MERIT results are averaged over five runs for MERIT + CASCADE decoder (Additive) + MUTATION. ↑ denotes higher the better, ↓ denotes lower the better. The best results are in bold.

| Architectures | Average DICE↑ | Average HD95[a]↓ | Aorta | GB[b] | KL[b] | KR[b] | Liver | PC[b] | SP[b] | SM[b] |
|---|---|---|---|---|---|---|---|---|---|---|
| UNet (Ronneberger et al., 2015) | 70.11 | 44.69 | 84.00 | 56.70 | 72.41 | 62.64 | 86.98 | 48.73 | 81.48 | 67.96 |
| AttnUNet (Oktay et al., 2018) | 71.70 | 34.47 | 82.61 | 61.94 | 76.07 | 70.42 | 87.54 | 46.70 | 80.67 | 67.66 |
| R50+UNet (Chen et al., 2021) | 74.68 | 36.87 | 84.18 | 62.84 | 79.19 | 71.29 | 93.35 | 48.23 | 84.41 | 73.92 |
| R50+AttnUNet (Chen et al., 2021) | 75.57 | 36.97 | 55.92 | 63.91 | 79.20 | 72.71 | 93.56 | 49.37 | 87.19 | 74.95 |
| SSFormerPVT (Wang et al., 2022b) | 78.01 | 25.72 | 82.78 | 63.74 | 80.72 | 78.11 | 93.53 | 61.53 | 87.07 | 76.61 |
| PolypPVT (Dong et al., 2021) | 78.08 | 25.61 | 82.34 | 66.14 | 81.21 | 73.78 | 94.37 | 59.34 | 88.05 | 79.4 |
| TransUNet (Chen et al., 2021) | 77.48 | 31.69 | 87.23 | 63.13 | 81.87 | 77.02 | 94.08 | 55.86 | 85.08 | 75.62 |
| SwinUNet (Cao et al., 2021) | 79.13 | 21.55 | 85.47 | 66.53 | 83.28 | 79.61 | 94.29 | 56.58 | 90.66 | 76.60 |
| MT-UNet (Wang et al., 2022a) | 78.59 | 26.59 | 87.92 | 64.99 | 81.47 | 77.29 | 93.06 | 59.46 | 87.75 | 76.81 |
| MISSFormer (Huang et al., 2021) | 81.96 | 18.20 | 86.99 | 68.65 | 85.21 | 82.00 | 94.41 | 65.67 | 91.92 | 80.81 |
| CASTformer (You et al., 2022) | 82.55 | 22.73 | **89.05** | 67.48 | 86.05 | 82.17 | **95.61** | 67.49 | 91.00 | 81.55 |
| PVT-CASCADE (Rahman and Marculescu, 2023) | 81.06 | 20.23 | 83.01 | 70.59 | 82.23 | 80.37 | 94.08 | 64.43 | 90.1 | 83.69 |
| TransCASCADE (Rahman and Marculescu, 2023) | 82.68 | 17.34 | 86.63 | 68.48 | 87.66 | 84.56 | 94.43 | 65.33 | 90.79 | 83.52 |
| Parallel MERIT (**Ours**) | 84.22 | 16.51 | 88.38 | 73.48 | 87.21 | 84.31 | 95.06 | 69.97 | 91.21 | 84.15 |
| Cascaded MERIT (**Ours**) | **84.90** | **13.22** | 87.71 | **74.40** | **87.79** | **84.85** | 95.26 | **71.81** | **92.01** | **85.38** |

[a] More details in Appendix B.2. [b] More details in Appendix B.1.

### 4.1. Results on Synapse multi-organ segmentation

Table 1 presents the results of Synapse multi-organ segmentation; it can be seen that both variants of our MERIT significantly outperform all the SOTA CNN- and transformer-based 2D medical image segmentation methods. Among all the methods, our Cascaded MERIT achieves the best average DICE score (84.90%). Cascaded MERIT outperforms two popular methods on this dataset, such as TransUNet and SwinUNet by 7.42% and 5.57%, respec-

Table 2: Results on the ACDC dataset. DICE scores (%) are reported for individual organs. We present the results of MERIT averaging over five runs with the setting MERIT + CASCADE decoder (Additive) + MUTATION. The best results are in bold.

| Architectures | Avg DICE | RV[a] | Myo[a] | LV[a] |
|---|---|---|---|---|
| R50+UNet (Chen et al., 2021) | 87.55 | 87.10 | 80.63 | 94.92 |
| R50+AttnUNet (Chen et al., 2021) | 86.75 | 87.58 | 79.20 | 93.47 |
| ViT+CUP (Chen et al., 2021) | 81.45 | 81.46 | 70.71 | 92.18 |
| R50+ViT+CUP (Chen et al., 2021) | 87.57 | 86.07 | 81.88 | 94.75 |
| TransUNet (Chen et al., 2021) | 89.71 | 88.86 | 84.53 | 95.73 |
| SwinUNet (Cao et al., 2021) | 90.00 | 88.55 | 85.62 | 95.83 |
| MT-UNet (Wang et al., 2022a) | 90.43 | 86.64 | 89.04 | 95.62 |
| MISSFormer (Huang et al., 2021) | 90.86 | 89.55 | 88.04 | 94.99 |
| PVT-CASCADE (Rahman and Marculescu, 2023) | 91.46 | 88.9 | 89.97 | 95.50 |
| TransCASCADE (Rahman and Marculescu, 2023) | 91.63 | 89.14 | **90.25** | 95.50 |
| Parallel MERIT (**Ours**) | **92.32** | **90.87** | 90.00 | **96.08** |
| Cascaded MERIT (**Ours**) | 91.85 | 90.23 | 89.53 | 95.80 |

[a] More details in Appendix B.1.

tively, when compared to their original reported DICE scores. Cascaded MERIT achieves 2.22% better DICE than the existing best method, TransCASCADE (82.68% DICE), on this dataset. When we compare the HD95 distance of all the methods, we find that both variants of our MERIT achieve a lower HD95 distance. Cascaded MERIT has the lowest HD95 distance (13.22) which is 18.47 lower than TransUNet (HD95 of 31.69) and 4.12 lower than the best SOTA method, TransCASCADE (HD95 of 17.34).

If we look into the DICE score of individual organs, we observe that proposed MERIT variants significantly outperform SOTA methods on six out of eight organs. We also can conclude that Cascaded MERIT performs better both in large and small organs, though it exhibits greater improvement for small organs. We believe that both MERIT variants demonstrate better performance due to using the multi-scale hierarchical transformer encoder with cascaded attention-based decoding and the MUTATION loss aggregation.

## 4.2. Results on ACDC cardiac organ segmentation

Table 2 reports three cardiac organ segmentation results of different methods on the ACDC dataset for MRI data modality. Both our Parallel and Cascaded MERIT have better DICE scores than all other SOTA methods. Our Parallel MERIT achieves the best average DICE score (92.32%) which outperforms TransUNet and SwinUNet by 2.61% and 2.32%, respectively. Parallel MERIT also shows the best DICE scores in RV[B.1] (90.87%) and LV[B.1] (96.08%) segmentation. We can conclude from these results that our method performs the best across different medical imaging data modalities.

## 5. Conclusion

In this paper, we have introduced a novel multi-scale hierarchical transformer architecture (MERIT) that can capture both the multi-scale and multi-resolution features necessary for accurate medical image segmentation. We have also incorporated an attention-based cascaded decoder to further refine features. Moreover, we have proposed a novel multi-stage feature mixing loss aggregation (MUTATION) strategy for implicit ensembling/augmentation which ensures better model training and boosts the performance without introducing additional hyper-parameters and inference overhead. Our experimental results on two well-known multi-class medical image segmentation benchmarks demonstrate the superiority of our proposed method over all SOTA approaches. Finally, we believe that our proposed MERIT architectures and MUTATION loss aggregation strategy will improve other downstream medical image segmentation and semantic segmentation tasks.

## Acknowledgments

This work is supported, in part, by NSF grant CNS 2007284.

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

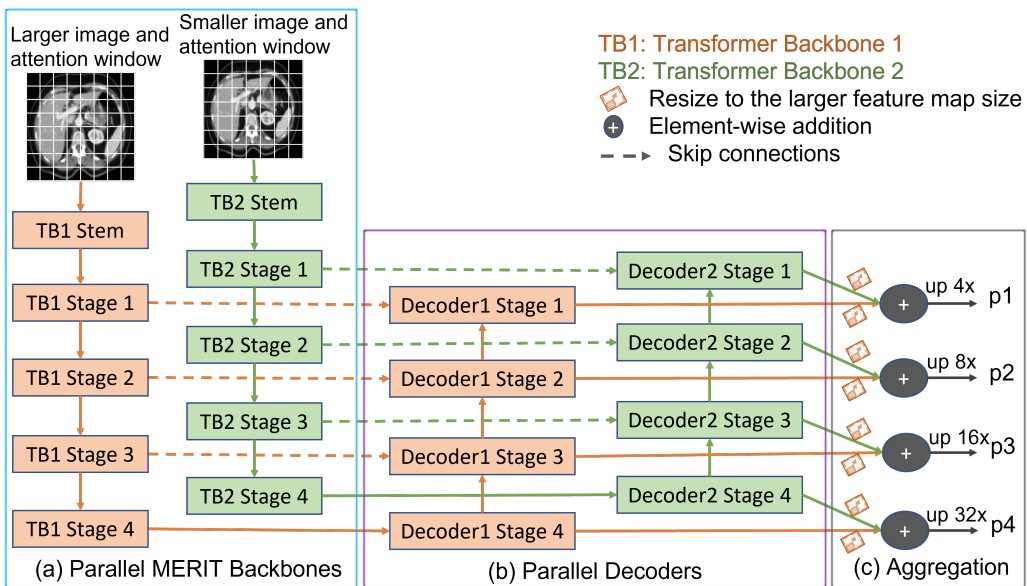

Figure 2: Parallel MERIT architecture. (a) Parallel MERIT backbone, (b) Parallel decoders, (c) Prediction maps aggregation from two decoders. p1, p2, p3, and p4 are the aggregated multi-stage prediction maps.

## Appendix A. Parallel MERIT Architecture

Our Parallel MERIT architecture is given in Fig. 2. This architecture is described in Section 3.1.2 of the main text.

## Appendix B. Experimental Setup

This section first describes datasets, then introduces evaluation metrics, and finally provides the implementation details of our proposed architecture and experiments.

### B.1. Datasets

**Synapse multi-organ dataset.** There are 30 abdominal CT scans with 3779 axial contrast-enhanced abdominal CT images in the Synapse multi-organ dataset[1]. Each CT scan has 85-198 slices of resolution $512 \times 512$ pixels, having a voxel spatial resolution of $([0:54\text{-}0:54] \times [0:98\text{-}0:98] \times [2:5\text{-}5:0]) mm^3$. We extract 2D slices from the CT scans and segment 8 abdominal organs, such as the aorta, gallbladder (GB), left kidney (KL), right kidney (KR), liver, pancreas (PC), spleen (SP), and stomach (SM). Following the experimental protocol of TransUNet (Chen et al., 2021), we split the dataset into 18 scans (2211 axial slices) for training, and 12 for validation.

**ACDC dataset.** The ACDC dataset[2] contains 100 cardiac MRI scans collected from different patients. We extract 2D slices from each MRI scan and segment three organs, such

---

1. https://www.synapse.org/#!Synapse:syn3193805/wiki/217789

2. https://www.creatis.insa-lyon.fr/Challenge/acdc/

as the right ventricle (RV), left ventricle (LV), and myocardium (Myo). Following MT-UNet (Wang et al., 2022a), we split the dataset into 70 (1304 axial slices), 10 (182 axial slices), and 20 cases for training, validation, and testing, respectively.

### B.2. Evaluation metrics.

In our experiments on the Synapse Multi-organ dataset, we use DICE and 95% Hausdorff Distance (HD95) as the evaluation metrics. However, we use only DICE scores as an evaluation metric for the ACDC dataset. The DICE similarity scores $DSC(Y, \hat{Y})$ and HD95 distance $D_H(Y, \hat{Y})$ (95th percentile of the distances between boundary points in $Y$ and $\hat{Y}$) are calculated using Equations 2 and 3, respectively.

$$DSC(Y, \hat{Y}) = \frac{2 \times |Y \cap \hat{Y}|}{|Y| + |\hat{Y}|} \times 100 \tag{2}$$

$$D_H(Y, \hat{Y}) = \max\{d_{Y\hat{Y}}, d_{\hat{Y}Y}\} = \max\{\max_{y \in Y} \min_{\hat{y} \in \hat{Y}} d(y, \hat{y}), \{\max_{\hat{y} \in \hat{Y}} \min_{y \in Y} d(y, \hat{y})\} \tag{3}$$

where $Y$ and $\hat{Y}$ are the ground truth mask and predicted segmentation map, respectively.

### B.3. Implementation details

We use PyTorch 1.12.0 with CUDA 11.6 in all of our experiments. Besides, we use a single NVIDIA RTX A6000 GPU with 48GB of memory to train all models. We utilize the Pytorch pre-trained weights on ImageNet from the *timm* library (Wightman, 2019) for MaxViT backbone networks. We use the input resolutions and attention windows of $\{(224 \times 224), (256 \times 256)\}$ and $\{(8 \times 8), (7 \times 7)\}$, respectively, in our (dual-scale) MERIT. We augment data using only random rotation and flipping. We train our model using AdamW optimizer (Loshchilov and Hutter, 2017) with a weight decay and learning rate of 0.0001. We optimize the combined DICE and Cross-Entropy (CE) loss $\mathcal{L}$ in Equation 4 with $\lambda_1 = 0.7$ and $\lambda_2 = (1 - \lambda_1) = 0.3$ (weights are selected empirically in Appendix C.7) in all our experiments:

$$\mathcal{L} = \lambda_1 \mathcal{L}_{DICE} + \lambda_2 \mathcal{L}_{CE} \tag{4}$$

where $\lambda_1$ and $\lambda_2$ are the weights for the DICE ($\mathcal{L}_{DICE}$) and CE ($\mathcal{L}_{CE}$) losses, respectively.

We train each model a maximum of 300 epochs with a batch size of 24 for Synapse multi-organ segmentation. For ACDC cardiac organ segmentation, we use a batch size of 12 and train each model for a maximum of 400 epochs.

## Appendix C. Ablation Studies

In this section, we present a wide range of ablation studies to answer different intrinsic questions related to our proposed architectures, loss aggregation, and experiments; these are described in the following subsections.

Table 3: Comparison with the baseline method on Synapse multi-organ and ACDC datasets. We report the results of our MERIT with the setting MERIT+CASCADE decoder(Additive)+MUTATION. We report the average inference time (ms) over 5000 samples. All reported DICE scores (%) in columns Synapse Multi-organ and ACDC are averaged over five runs. The best results are in bold.

| Architectures | Input Resolutions | Params (M)/ FLOPS (G) | Inference Time (ms) | Synapse Multi-organ | ACDC |
|---|---|---|---|---|---|
| MaxViT | $224 \times 224$ | 65.25/10.43 | 19.79 | 77.11 | 90.56 |
| MaxViT | $256 \times 256$ | 65.25/14.19 | 20.58 | 78.53 | 90.98 |
| MaxViT with CASCADE decoder | $224 \times 224$ | 82.62/14.2 | 21.84 | 79.83 | 90.87 |
| MaxViT with CASCADE decoder | $256 \times 256$ | 82.62/19.11 | 23.07 | 80.20 | 91.15 |
| Parallel MERIT (**Ours**) | $256 \times 256$, $224 \times 224$ | 147.86/33.31 | 37.01 | 84.22 | **92.32** |
| Cascaded MERIT (**Ours**) | $256 \times 256$, $224 \times 224$ | 147.86/33.31 | 37.06 | **84.90** | 91.85 |

### C.1. Comparison with the baseline method

We compare our proposed methods with baseline hierarchical MaxViT architecture. In the case of MaxViT, we do the same multi-stage prediction for a fair comparison. We also use a similar experimental setting except using MUTATION with our architectures. Table 3 presents the results of these experiments. We can see from Table 3 that our proposed architectures with MUTATION loss (see "our" entries in Table 3) improve the baseline hierarchical $256 \times 256$ resolution MaxViT (see $2^{nd}$ row entries in Table 3) by 6.37% and 1.34% DICE scores (with $1.62 \times$ more FLOPS and $1.8 \times$ longer inference time) in Synapse multi-organ and ACDC datasets, respectively. We can also see that our MERIT architecture has 147.86M parameters which is $1.8 \times$ larger than $256 \times 256$ resolution MaxViT with CASCADE decoder (see $4^{th}$ row entries in Table 3), but with a 4.7% better DICE score in Synapse multi-organ. We think that this increase in parameters/FLOPS/inference time is worth it given the improvement in performance.

### C.2. Effect of multi-scale backbone

We have conducted experiments on the Synapse multi-organ dataset to show the effect of our multi-scale backbone on medical image segmentation. In Table 4, we present the results of all the methods with the CASCADE decoder (no MUTATION) to make a fair comparison of our proposed architecture. It can be seen from Table 4 that the input resolution has an impact on DICE score improvement. More precisely, $256 \times 256$ resolution backbones have better DICE scores than the $224 \times 224$ resolution backbones. As shown, our Cascaded MERIT achieves the best DICE score (83.35%) which improves the baseline $256 \times 256$ resolution MaxViT (see $2^{nd}$ row entries in Table 4) by 3.15%. When comparing with the double backbone architectures with the same input scale (attention window), we can see that our multi-scale (attention window) double backbone architectures achieve better DICE scores due to their additive advantage of multi-scale feature extraction. We note that our Paral-

Table 4: Effect of multi-scale backbone on Synapse multi-organ dataset. We report the results of the backbone with CASCADE decoder (no MUTATION) to clarify the effect of multi-scale backbones. All reported results are averaged over five runs. The best results are in bold.

| Architectures | Input Resolutions | Attention Windows | Params (M)/ FLOPS (G) | Avg DICE (%) |
|---|---|---|---|---|
| (single) MaxViT | $224 \times 224$ | $7 \times 7$ | 82.62/14.2 | 79.83 |
| (single) MaxViT | $256 \times 256$ | $8 \times 8$ | 82.62/19.11 | 80.20 |
| Parallel Double MaxViT | $224 \times 224, 224 \times 224$ | $7 \times 7, 7 \times 7$ | 147.86/28.4 | 80.81 |
| Parallel Double MaxViT | $256 \times 256, 256 \times 256$ | $8 \times 8, 8 \times 8$ | 147.86/38.22 | 82.15 |
| Cascaded Double MaxViT | $224 \times 224, 224 \times 224$ | $7 \times 7, 7 \times 7$ | 147.86/28.4 | 81.06 |
| Cascaded Double MaxViT | $256 \times 256, 256 \times 256$ | $8 \times 8, 8 \times 8$ | 147.86/38.22 | 83.02 |
| Parallel MERIT (**Ours**) | $256 \times 256, 224 \times 224$ | $8 \times 8, 7 \times 7$ | 147.86/33.31 | **82.91** |
| Cascaded MERIT (**Ours**) | $256 \times 256, 224 \times 224$ | $8 \times 8, 7 \times 7$ | 147.86/33.31 | **83.35** |

Table 5: Comparison of Tiny MERIT vs. Small MaxViT architectures on Synapse multi-organ dataset. We report the results of the backbone with CASCADE decoder (no MUTATION) to clarify the effect of multi-scale backbones. All reported results are averaged over five runs. The best results are in bold.

| Architectures | Input Resolution | Attention Windows | Params (M)/ FLOPS (G) | Avg DICE (%) |
|---|---|---|---|---|
| MaxViT-Tiny | $224 \times 224$ | $7 \times 7$ | 36.86/6.57 | 77.84 |
| MaxViT-Tiny | $256 \times 256$ | $8 \times 8$ | 36.86/8.61 | 78.43 |
| Parallel MERIT-Tiny (**Ours**) | $256 \times 256, 224 \times 224$ | $8 \times 8, 7 \times 7$ | 65.41/15.18 | **81.34** |
| Cascaded MERIT-Tiny (**Ours**) | $256 \times 256, 224 \times 224$ | $8 \times 8, 7 \times 7$ | 65.41/15.18 | **81.82** |
| MaxViT-Small | $224 \times 224$ | $7 \times 7$ | 82.62/14.2 | 79.83 |
| MaxViT-Small | $256 \times 256$ | $8 \times 8$ | 82.62/19.11 | 80.20 |

lel/Cascaded MERIT (33.31G) has a significantly lower computational complexity/FLOPS than the $256 \times 256$ resolution Parallel/Cascaded Double MaxViT (38.22G) due to using one $256 \times 256$ and another $224 \times 224$ resolution inputs. Despite that, our Parallel and Cascaded MERIT outperform the Double MaxViT by 0.76% and 0.33%, respectively. These improvements in the DICE score support the claim regarding the benefit of calculating SA in multiple scale attention windows.

We have conducted an additional set of experiments by implementing a tiny version of MERIT using the tiny MaxViT backbones, to clarify that performance improvement is due to the effect of multi-scale SA, not because of using a model with more parameters. As

Table 6: Effect of CASCADE decoder and MUTATION loss aggregation in MERIT on Synapse multi-organ dataset. We present the results of MERIT averaging over five runs. The best results are in bold.

| Architectures | CASCADE decoder | MUTATION | Avg DICE (%) |
|---|---|---|---|
| Parallel MERIT | No | No | 80.44 |
| Parallel MERIT | No | Yes | 81.06 |
| Parallel MERIT | Yes | No | 82.91 |
| Parallel MERIT (**Ours**) | Yes | Yes | 84.22 |
| Cascaded MERIT | No | No | 80.76 |
| Cascaded MERIT | No | Yes | 82.03 |
| Cascaded MERIT | Yes | No | 83.35 |
| Cascaded MERIT (**Ours**) | Yes | Yes | **84.90** |

shown in Table 5, when comparing against the Small MaxViT backbone which has more model parameters, both our Tiny MERIT backbones perform better. Our Tiny Cascaded MERIT backbone outperforms the Small MaxViT backbone (see $4^{th}$ row entries in Table 5) by up to 1.62% DICE score for a $256 \times 256$ input resolution while having $1.26 \times$ smaller model parameters and $1.26 \times$ fewer FLOPS. Therefore, again, we can conclude from these empirical evaluations that our multi-scale SA calculation improves the performance of medical image segmentation.

### C.3. Effect of CASCADE decoder and MUTATION loss aggregation in MERIT

We have conducted some experiments on Synapse multi-organ dataset to demonstrate the effect of CASCADE decoder and MUTATION loss aggregation strategy on our MERIT architectures. Table 6 presents the results of our Parallel MERIT with or without CAS-CADE decoder and MUTATION. We can see from Table 6 that Parallel and Cascaded MERIT without both CASCADE decoder and MUTATION have the lowest DICE scores. CASCADE decoder significantly increases the DICE scores (2.47-2.59%) due to capturing the spatial (contextual) relations among pixels (usually limited in vision transformer), while MUTATION alone marginally improves the DICE (0.62-1.27%). However, when MUTA-TION is used with the outputs from CASCADE decoder, it achieves the best DICE scores (84.22%, 84.90%) improving CASCADE decoder by 1.31-1.55%. We believe the reason behind this is that MUTATION works well with the refined features of the CASCADE decoder. Therefore, we can conclude that the synthesized prediction maps generated via combinatory aggregation (MUTATION) help us improve the performance of the model; this is why we prefer combinatory loss aggregation over linear aggregation. We believe that our combinatory loss aggregation (MUTATION) can be used as a beneficial ensembling/augmentation method in other downstream semantic and medical image segmentation tasks.

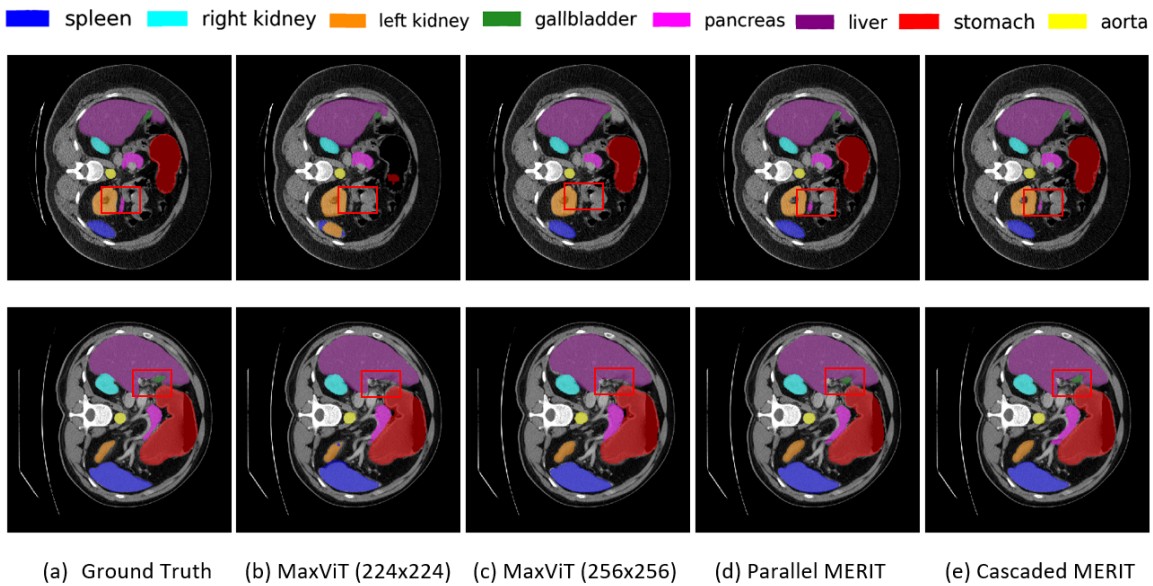

(a) Ground Truth     (b) MaxViT (224x224)    (c) MaxViT (256x256)    (d) Parallel MERIT     (e) Cascaded MERIT

Figure 3: Qualitative results on Synapse multi-organ dataset. (a) Ground Truth (GT), (b) MaxViT (input resolution 224×224), (c) MaxViT (input resolution 256×256), (d) Parallel MERIT, (e) Cascaded MERIT. We produce all the segmentation maps with the CASCADE decoder and overlay on top of original image/slice.

Table 7: Comparison of different aggregations in CASCADE decoder on Synapse multi-organ dataset. We present the results of MERIT averaging over five runs with the setting MERIT + CASCADE decoder + MUTATION. The best results are in bold.

| Architectures | Aggregation in CASCADE decoder | Avg DICE (%) |
|---|---|---|
| Parallel MERIT | Concatenation | 84.18 |
| Parallel MERIT | Concatenation | 84.22 |
| Cascaded MERIT | Additive | 84.88 |
| Cascaded MERIT | Additive | **84.90** |

## C.4. Qualitative results on Synapse Multi-organ Segmentation

Fig. 3 shows the qualitative results of the baseline hierarchical MaxViT and our proposed MERIT architectures. As shown in the figure, our MERIT architecture can segment the small organs (see the red rectangular box) well. In contrast, the single scale MaxViT architecture with both 224 × 224 and 256 × 256 input resolutions fail to segment that small organ. Our MERIT architecture also segments the larger organ much better than the single

Table 8: Comparison of different interpolations in MERIT on Synapse multi-organ dataset. We present the results of MERIT averaging over five runs with the setting MERIT + CASCADE decoder(Additive) + MUTATION. The best results are in bold.

| Architectures | Interpolations | Avg DICE (%) |
|---|---|---|
| Parallel MERIT | nearest-exact | 81.67 |
| Parallel MERIT | area | 81.76 |
| Parallel MERIT | bicubic | 83.58 |
| Parallel MERIT | bilinear | 84.22 |
| Cascaded MERIT | nearest-exact | 82.27 |
| Cascaded MERIT | area | 82.38 |
| Cascaded MERIT | bicubic | 84.05 |
| Cascaded MERIT | bilinear | **84.90** |

scale MaxViT. We believe the reason behind this better segmentation of both small and large organs by our MERIT architectures is the use of multi-scale SA.

### C.5. Effect of different aggregations in CASCADE decoder

Table 7 presents the results of concatenation and additive aggregations in CASCADE decoder on Synapse multi-organ dataset. We can see from Table 7 that our MERIT architectures with additive aggregation in CASCADE decoder are marginally better (0.04-0.02%) than the concatenation. Therefore, we can conclude from these results that the aggregation techniques do not have much impact on the CASCADE decoder of our architectures while using MUTATION. However, concatenation aggregation-based methods usually have additional computational overheads due to increasing the number of channels after the aggregation, while additive aggregation keeps the number of channels the same. Consequently, we recommend using additive aggregation in our MERIT architectures due to its computational benefits.

### C.6. Effect of different interpolations in MERIT

We have conducted some experiments on Synapse multi-organ dataset to choose the best interpolations methods for our proposed MERIT architectures. Table 8 presents the results of Parallel and Cascaded MERIT using *nearest-exact* (nearest neighbor), *area*, *bicubic*, and *bilinear* interpolation methods from Pytorch. The *nearest-exact* interpolation shows the lowest DICE scores while *bilinear* and *bicubic* interpolation achieve the best and second best DICE scores, respectively. Therefore, we recommend using *bilinear* interpolation in our proposed MERIT architectures to re-scale the features and prediction maps.

### C.7. Choosing weight for DICE and CE losses

We optimize the combined DICE and CE loss during the training of our models. Here, we have conducted some experiments to choose the best weight pairs to combine these two

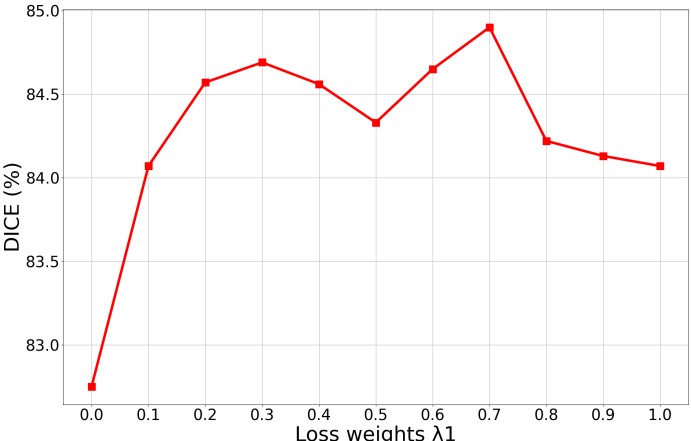

Figure 4: Loss weight vs. DICE curve on Synapse multi-organ dataset. We report the results of our Cascaded MERIT+CASCADE decoder(Additive) + MUTATION with different weights for DICE and CE losses. X-axis presents the weights of DICE loss, $\lambda_1$, while the weight for CE loss is $\lambda_2 = 1 - \lambda_1$. The value of 0.0 on the X-axis represents weights for DICE and CE losses of 0.0 and 1.0, respectively (i.e., only CE loss is used). While the value of 1.0 on the X-axis represents weights for DICE and CE losses of 1.0 and 0.0, respectively (i.e., only DICE loss is used)

losses. Fig. 4 presents the DICE scores for different weight pairs for losses. We can see in the graph that the model shows the worst DICE score when using only the CE loss. We get the best DICE score for the weights pair $(\lambda_1, \lambda_2) = (0.7, 0.3)$ which we have used in all of our experiments.

## Appendix D. Supplementary Materials

We make our source code publicly available at $https://github.com/SLDGroup/MERIT$.

