# OpenReview forum: "Multi-scale Hierarchical Vision Transformer with Cascaded Attention Decoding for Medical Image Segmentation"
_MIDL.io/2023/Conference — MIDL 2023 Poster_

### Official Review · Reviewer_o4gS · 2023-02-03

**Confidence:** 4
**Preliminary Rating:** 3
**Recommendation:** Poster

**Summary:**

In this paper, the authors propose a new transformer-based network architecture for medical image segmentation. The architecture consists of a multi-scale hierarchical encoder and a cascaded attention decoder. Besides, a multi-stage feature-mixing loss is also proposed. The performance of the proposed method is evaluated on two datasets: synapse and ACDC, which have been widely used for benchmarking the performance of emerging network architectures for medical image segmentation tasks. Experimental results show that the proposed method achieves state-of-the-art performance compared to existing methods.

**Strengths:**

- The paper is well written in terms of method description and experiment design.
- The proposed approach achieved the state-of-the-art performance on two widely used datasets. If the source code are publicly available and the experimental results can be reproduced, the proposed method should have a positive impact to the field.

**Weaknesses:**

- The novelty of the proposed method is a major concern. The cascaded attention decoder and the multi-stage feature aggregation loss have been previously proposed in [1]. The differences between this method and [1] are (1) increase the number of encoder from 1 to 2 (parallel or cascaded), and (2) modify the multi-stage loss by allowing loss computation on different combination of the outputs of prediction heads.

- The motivation of using dual encoder is not very clear and the design itself suffers from computational complexity (doubling the parameters as using single encoder), which at least should be discussed in the paper. The effectiveness of using dual encoders is hard to be validated because the performance gain could be simply caused by using a larger network (more parameters) instead of multi-scale features.

- For 3D segmentation tasks, nnU-Net [2] should be compared as it has achieved state-of-the-art performances in many segmentation challenges.


[1] Rahman, M. M., & Marculescu, R. (2023). Medical Image Segmentation via Cascaded Attention Decoding. In Proceedings of the IEEE/CVF Winter Conference on Applications of Computer Vision (pp. 6222-6231).
[2] Isensee, F., Jaeger, P. F., Kohl, S. A., Petersen, J., & Maier-Hein, K. H. (2021). nnU-Net: a self-configuring method for deep learning-based biomedical image segmentation. Nature methods, 18(2), 203-211.

**Deanonymize Review:**

no

**Detailed Comments:**

- As mentioned in the weakness section, the benefit of dual encoder should be explained clearly (why the performance gain is from multi-scale features instead of larger network).
- The computational complexity needs to be discussed.
- Comparison to nnU-Net should be added.

**Paper Type:**

methodological development

**Questions To Address In The Rebuttal:**

- As mentioned in detailed comments, the benefit of dual encoder should be explained clearly (why the performance gain is from multi-scale features instead of larger network).

- Please discuss the model parameters and inference time for compared methods.

---

### Official Review · Reviewer_Y7cf · 2023-02-04

**Confidence:** 4
**Preliminary Rating:** 2

**Summary:**


This work presents two vision transformer based architectures for 2D segmentation that aim to consider multi-scale and multi-resolution features in medical imaging data. The key contributions are in employing two backbone architectures to capture multi-scale/multi-resolution features. The work also introduces a new loss aggregation strategy that uses intermediate predictions using features for improving training behaviour. Experiments on two datasets show competitive performance compared to relevant baseline methods.


**Strengths:**

* Use of multiple backbones to capture multi-scale/multi-resolution features is reasonable.

* Ablation studies reported in the Appendix are quite insightful, and provide useful peek into the roles of the different modules used in this work.

* The performance of the proposed methods (cascaded and parallel) shows improvements compared to the baseline methods, in most cases and in both datasets

* The literature review is fairly detailed (although misses couple of relevant papers from MIDL2022) and the paper is largely clearly written

* Authors promise to make the trained model available with proper documentation

**Weaknesses:**

* **Limitations described**: First limitation in Page 2 states:
> 1) self-attention is performed with a single attention window (scale) which has limited feature processing ability

Is this true when discussing other hierarchical transformer models? Pyramidal ViT, Swin and Max ViT do operate at different scales. Are these models not using attention at multiple scales in some form or another?

* **Feature-mixing loss**: The ablation studies show that the feature-mixing loss do help the performance of the models. What is the underlying mechanism for these aggregated predictions to help with the training/convergence of the models? Is this a form of implicit ensembling?

* **Difference in  cascaded and parallel MERIT**: In what scenarios do the authors recommend using the parallel model compared to the cascaded one. Each model fares better than the other on one dataset. Seen from a segmentation point of view, what are the advantages of parallel and cascade models?

* **Performance comparisons**: The experiments and results are detailed. However, the model complexity of different methods is missing. In some sense, the proposed model is easily twice as complex as the MaxViT model. Reporting of the number of parameters, training time, energy consumption or similar measures could provide more insight into the trade-offs.

* **Validation of claims**: The two limitations listed in Page 2 about self-attention having limited feature processing ability and limited ability to learn spatial relations are not explicitly validated. Is the performance improvement due to these reasons or some other improvements? As there are no qualitative examples or analysis of segmentation errors, it is difficult to tie the improvements in Dice as an indicator for solving these issues.

**Deanonymize Review:**

no

**Detailed Comments:**

See comments above.

**Paper Type:**

both

**Questions To Address In The Rebuttal:**

The most important points that can be addressed are about:
* Reporting model complexity to provide a more balanced view of the performance
* Motivating when to use parallel and cascade models
* A discussion on how the proposed models are alleviating the limitations identified in other hierarchical ViT models

---

### Official Review · Reviewer_cGNY · 2023-02-07

**Confidence:** 5
**Preliminary Rating:** 4
**Recommendation:** Oral, Poster

**Summary:**

The authors introduced a multi-scale hierarchical vision transformer for 2D medical segmentation, with a cascaded attention-based decoder and multi-stage feature-mixing loss aggregation during loss calculation. The validation was performed on Synapse multi-organ segmentation and ACD cardiac organ segmentation with significant improvement on average DICE.

**Strengths:**

The paper is easy to follow. The multi-stage multi-scale method looks interesting. The loss aggregation has the potential to be generalized to any multi-stage segmentation or dense prediction networks. The proposed method was compared with many recent works' baseline methods, including UNet-based and Transformer based.

**Weaknesses:**

The paper is well written. My minor concern is why combinatory loss aggregation is preferred over linear aggregation.

There are a few technical details (especially on network structures) that are not clear in the main manuscript. Please check the 'Questions To Address In The Rebuttal' section.

**Deanonymize Review:**

no

**Detailed Comments:**

The ablation study results need to be discussed in the main manuscripts. For the clarification issue, please check the 'Questions To Address In The Rebuttal' section.

**Paper Type:**

both

**Questions To Address In The Rebuttal:**

- referenced from Figure 1, why is OC-S not counted in Decoder2?
- a follow-up question, why the merged features only are only sent to Decoder1 rather than Decoder2?
- why there is no OC-S module in parallel MERIT mode?
- For algorithm 1, what is the definition of loss_function()?

- page 4, 'we also resize the feature map to the smaller input resolution', please specify.

---

### Meta-Review · Area_Chair_nEYy · 2023-02-19

**Recommendation:** Accept (Poster)
**Confidence:** 4

**Metareview:**

While the reviewers appreciated the comprehensiveness of the evaluation and high writing quality, the reviewers expressed their concerns about the detailed methodology. These include the lack of illustrations for details and the lack of innovation. In the first round of the review, the reviewers gave “weak accept”, “weak reject”, and “borderline”. The reviewers did not change their ratings after seeing the rebuttal. After checking all rebuttals, I think the concerns have been largely addressed. Thus, my recommendation leans toward acceptance.